# Synergistic Action of Stilbenes in Muscadine Grape Berry Extract Shows Better Cytotoxic Potential Against Cancer Cells Than Resveratrol Alone

**DOI:** 10.3390/biomedicines7040096

**Published:** 2019-12-05

**Authors:** Subramani Paranthaman Balasubramani, Mohammad Atikur Rahman, Sheikh Mehboob Basha

**Affiliations:** Center for Viticulture and Small Fruit Research, Florida Agricultural and Mechanical University, 6361, Mahan Drive, Tallahassee, FL 32317, USA; balasubra.paranthama@famu.edu (S.P.B.); mohammad.rahman@famu.edu (M.A.R.)

**Keywords:** muscadine grape, anti-cancer, resveratrol, stilbenes, synergism

## Abstract

Muscadine grape is rich in stilbenes, which include resveratrol, piceid, viniferin, pterostilbene, etc. Resveratrol has been extensively studied for its biological activities; however, the synergistic effect of stilbene compounds in berry extracts is poorly understood. The aim of this study was to evaluate the anti-cancer activity of stilbene-rich muscadine berry extract and pure resveratrol. Stilbenes were extracted from ripened berries of muscadine grape cultivars, Pineapple, and Southern Home. HPLC analysis was performed to determine quantity of stilbenes. The extracts were tested for their cytotoxic activity against A549 (lung carcinoma cells), triple negative breast cancer (HCC-1806) and HepG2 (human liver cancer) cells. The stilbene-rich extracts of the muscadine berry extracts showed cytotoxic activity against all of the cells tested. The extracts at 1 μg/mL induced death in 50–80% of cells by 72 h of treatment. About 50 μg/mL of resveratrol was required to induce a similar response in the cells. Further, modulation of genes involved in tumor progression and suppression was significantly (*p* < 0.0005) higher with the HepG2 cells treated with stilbene-rich berry extracts than the pure resveratrol. This shows that the synergistic activity of stilbenes present in muscadine grape berries have more potent anti-cancer activity than the resveratrol alone.

## 1. Introduction

Muscadine grape (*Vitis rotundifolia* Michx.) is one of the important native fruit crops in southeastern United States of America (USA), and was recently introduced in China [1]. The berries are used commercially to manufacture wine, juice, and preserves. To date, more than 100 different cultivars are being grown and they produce berries differing in size, color, and flavor [2]. The unique characteristic of muscadine grape berry is its thick skin [3].

The berry has been extensively studied for its phytochemical composition. The presence of polyphenolics, anthocyanins, stilbenes, ellagic acid, ellagitannins, organic acids, amino acids, and sugars has been reported for berry extracts [4,5]. Antioxidant, anti-microbial [4], anti-cancer [6], anti-diabetic [7], anti-inflammatory [8], immunomodulatory [9], cardioprotective [10], anti-obesity [11], and inhibition of lipid accumulation [12] properties of muscadine berries have been studied using in vitro (cell-free and cell lines), in vivo, and human trials. Berry extracts have been shown to be effective in controlling the proliferation of blood [13], colon [14], prostate [15], and breast [6] cancers. Whole berry, skin, or seed extracts rich in phenolics, including ellagic acid, have been linked to their anti-cancer effect [6,13,14,15]. However, studies have not tried to identify and map the exact phytochemicals responsible for the biological activity.

Stilbenes are a major class of bioactive phenolic compounds present in grapes. The presence of common stilbenes like t-piceid, t-resveratrol, ε-viniferin, and t-pterostilbene in muscadine grape berries has been reported [16]. Resveratrol and other stilbenes have been extensively studied for their biological activities, including anti-cancer, anti-inflammatory, antioxidant activity, etc. [17]. Generally, the stilbenes are isolated in groups, but publications have failed to address and understand their synergistic action.

The aim of the current study was to do a comparative evaluation of the stilbene-rich muscadine berry extracts of selected cultivars and pure resveratrol for their anti-cancer activity in pulmonary, breast, and liver cancer cell lines.

## 2. Experimental Section

### 2.1. Chemicals

Analytical grade methanol, acetonitrile, ethyl acetate, trans-resveratrol, trans-piceid, trans-pterostilbene, and ε-viniferin were purchased from Sigma-Aldrich (St. Louis, MO, USA).

### 2.2. Berry Samples

Berries from two different cultivars of muscadine grapes (Pineapple and Southern Home) were collected from the vineyard at the Center for Viticulture and Small Fruit Research, Florida Agriculture and Mechanical University, Tallahassee, FL, USA (30.47 latitude and 84.17 longitude) during vintage of 2018 (August and September). Ripe berries (EL-38 stage) [18] were collected from different vines.

### 2.3. Extraction of Stilbenoids

The stilbenoids were extracted according to the described protocol [19], with slight modifications. Ten grams (10 g) of berry was ground with liquid nitrogen and the ground powder was mixed with 7 mL of methanol, and then homogenized for 10 min on a vortex and further extracted for 24 h under at darkness. The suspension was centrifuged at 14,000 rpm for 15 min. The supernatant was removed carefully, and the resulting residue was extracted a second time with 3 mL of methanol and ethyl acetate (1:1, *v*/*v*), as described above. The pooled organic solvent extracts were vacuum-dried in a concentrator (SpeedVac concentrator, Savant Instruments, Inc. Holbrook, NY, USA) at room temperature in the dark. The dried samples were then re-dissolved in methanol (1 mL) for HPLC studies and DMSO for bioactivity studies.

#### HPLC Analysis of Stilbenoid Content

The vacuum concentrated extracts of the berries were dissolved in 1 mL of HPLC grade methanol and filtered through a 0.22 µm nylon membrane filter (Sartorius Stedim Biotech GmbH 37070 Gottingen Germany, Goettingen, Germany). HPLC analysis of stilbenoids was carried out with a system equipped with a 2487 dual UV detector and 1525 gradient pump (Waters Corporation, Milford, MA, USA). The HPLC pumps, autosampler, and detectors were controlled via Waters Empower software (Empower 3 service pack 2) supplied by Waters Corporation (Milford, MA, USA). The analytical column Luna RP C18 (4.6 × 250 mm; particle size, 5 μm) and guard cartridge (C18 4 × 3.0 mm) were obtained from Phenomenex (Torrance, CA, USA). The column temperature was maintained at 25 °C. The gradient elution profile was as follows: 90% solvent water (B), 10% solvent acetonitrile (A) (0–18 min); 85% A, 15% B (18–23 min); 85% A 15% B (23–30 min); 10% A, 90% B (30–35 min). The flow rate was set at 0.4 mL/min. A volume of 2 µL of each sample was injected to resolve and measure individual stilbenoids. Three injections were performed in sequence for each biological replicate. UV absorbance detection was recorded using dual wavelengths at 285 and 305 nm. A mixture of the standards was prepared using 1, 2, 3, 4, 5, and 6 ng of each of the four stilbenoids and used for calibration and quantification of stilbenoids. The standards, t-piceid, t-resveratrol, ε-viniferin, and t-pterostilbene (Sigma Aldrich, St. Louis, MO, USA) were prepared and used as described above. Samples and calibration standards were run in triplicates. Chromatograms were acquired with different retention times for each of the stilbenes and the area under the curve (AUC) was calculated using Empower III software (Waters Corporation, Milford, MA, USA). The linearity ranges of the calibration curves were R^2^ = 0.9906. Quantification of stilbenoids from the two muscadine cultivars was based on the calibration curves obtained from the respective standards.

### 2.4. Cell Culture and Reagents

The A549 (human lung carcinoma), HCC1806 (human triple negative breast cancer) and HepG2 (human hepatocellular carcinoma) cell lines were directly obtained from ATCC (Manassas, VA, USA) for this study. The A549, HCC1806 cell lines, and HepG2 cells were cultured in RPMI1640 basal medium supplemented with 10% (*v*/*v*) fetal bovine serum, 100 U/mL penicillin, and 100 mg/mL streptomycin. All of the cells were grown at 37 °C in a humidified incubator containing 5% CO_2_. Cell growth and viability were determined using a Bio-Rad TC-20 automated cell counter (Hercules, CA, USA).

#### 2.4.1. Cytotoxicity Assay

The cytotoxicity assay was performed according to the method by Mosmann [20] with modifications. Confluent cells were seeded into three 96-well plates at 1 × 10^5^ cells/mL and incubated for 24 h in a humidified atmosphere of 5% CO_2_ at 37 °C. Cells were treated with serum-free medium containing varying concentrations of extracts dissolved in DMSO. The volume of DMSO was maintained not to exceed 2% *v*/*v* per well, and a DMSO treatment group was also maintained as control. Cells were then incubated in a humidified atmosphere of 5% CO_2_ at 37 °C. Cell viability was assessed using the Bio-Rad TC-20 automated cell counter at 24, 48, and 72 h. Results are expressed as percentage cell viability in comparison to the untreated control.

#### 2.4.2. Total RNA Extraction and cDNA Synthesis

The TRI Reagent-based method was used for RNA extraction [21]. In brief, 24 h extract (1 µg/mL) and resveratrol (50 µg/mL)-treated HepG2 cells were released from the substratum by scraping and by repeated pipetting. The cell suspension was centrifuged at 12,000 rpm for 10 min at 4 °C. Cells from different replicates were pooled to obtain about 3–4 × 10^6^ cells. To the pellet, 1 mL of TRI Reagent (Sigma-Aldrich, St. Louis, MO, USA) was added by repeated pipetting. After 5 min at room temperature, 0.2 mL of chloroform was added and vigorously shaken for 15 s. The mixture was allowed to stand at room temperature for 15 min and centrifuged at 12,000 rpm for 10 min at 4 °C. The upper aqueous phase was transferred to a fresh tube and 0.5 mL of ice-cold iso-propanol was added and allowed to stay at room temperature for 10 min. RNA was pelleted by centrifugation at 12,000 rpm for 10 min at 4 °C. The pellet was washed with 1 mL of 75% ethanol, dried, and dissolved in DEPC (Diethyl pyrocarbonate; 0.1%)-treated water. RNA was stored at −80 °C until use. The quantity and quality of RNA was checked using a nano-quantity spectrophotometer (Thermo Fisher Scientific, Waltham, MA, USA). cDNA was synthesized using the SuperScript III First-Strand Synthesis System (Invitrogen, Carlsbad, CA, USA), using oligo(dt) according to the manufacturer′s instruction, using 5 µg of the total RNA.

#### 2.4.3. Semi-Quantitative PCR

Semi-quantitative analysis of gene expression was assessed following the protocol described by Marone et al. [22]. Briefly, the cDNA (2 µL) was amplified with gene specific primers (0.3 µM) in a PCR mix composed of 50 mM KCl, 10 mM Tris-HCl pH 9.0, 1.5 mM MgCl_2_, 200 µM each dNTP and 2 Taq DNA polymerase, in a 10 μL final volume reaction. The reaction conditions were as follows: Initial denaturation at 95 °C for 5 min, followed by 95 °C for 30 s, 60 °C for 30 s, 72 °C for 45 s, and a final extension at 72 °C for 5 min in a thermal cycler (Eppendorf, Master Cycler Gradient, Hamburg, Germany). The amplicons were resolved alongside standard 100 bp DNA marker in a 2% agarose gel prepared with tris-acetic acid EDTA (TAE) buffer and documented using gel documentation system (Bio-Rad, Hercules, CA, USA). The bands were analyzed using the quantity and density analysis tools of Quantity One software (version 4.6.3). *β-actin* and *GAPDH* were used as standards to normalize the expression of the genes of interest.

### 2.5. Statistical Analysis

Data are presented as means ± standard deviation. Comparison of means was performed by Student′s *t*-test. A 95% level of confidence was considered; thus, *p* < 0.05 referred to statistical significance.

## 3. Results

The ethanol and ethyl acetate extract of berries showed the presence of stilbenes by HPLC analysis. The extracts possessed varied levels of cytotoxicity against the cancer cells tested. The stilbene-rich extracts showed better cytotoxic activity against the cancer cells than the resveratrol alone, by modulating cancer suppressor and promotor genes.

### 3.1. Extraction and Quantification of Stilbenes

The two cultivars selected for this study are morphologically and genetically different. Though there was no significant difference in the size of the berries produced by the selected varieties, Pineapple produces bronze-colored berries, while black-colored berries are found in Southern Home cultivar. The phytochemical composition and biological activities of these cultivars have not been studied earlier. Successive extraction with methanol and methanol: ethyl acetate mixture (1:1, *v*/*v*) extracted the common stilbenes from the berries of the cultivars. The HPLC protocol followed resolved t-piceid, t-resvetarol, Ɛ-viniferin, and t-pterostilbene with distinct retention times (Figure 1). The solvent system and the HPLC conditions followed produced reproducible profiles on repeats. The total stilbene content was found to be significantly higher in the cultivar Southern Home than Pineapple. Southern Home was found to have 2-fold higher t-piceid and about 50-fold higher Ɛ-Viniferin (Table 1). While t-pterostilbene was not detected in Pineapple, about 4.5 ± 0.1 µg/g of whole berry was present in Southern Home. The cultivars selected did not show much variation in their resveratrol content.

### 3.2. Stilbene-Rich Muscadine Berry Extracts Induce Cytotoxicity in Cancer Cells

The stilbene-rich extracts of muscadine grape extracts exhibited cytotoxicity against the cancer cells tested. The HepG2 liver cancer cells were reduced to less than half by 750 and 1000 ng/mL extracts of both Pineapple and Southern Home cultivars by 72 h (Figure 2). The extracts of both cultivars were found to have similar cytotoxicity patterns on these cells.

In the triple negative breast cancer cells (HCC1806), the Pineapple cultivar extract was found to be more effective than the Southern Home. More than 50% of the cells were killed by the Pineapple cultivar extract at 72 h with 500 ng/mL concentration of the extract (Figure 3a). At the higher concentration tested (1000 ng/mL), only 38% of cells survived at 72 h. The Southern Home extract was effective in reducing the cells to 50% only at the higher concentration tested (1000 ng/mL) by 72 h (Figure 3b).

The stilbene-rich muscadine grape extracts were able to reduce the percentage of surviving pulmonary cancer cells (A549) by half, with 250 and 500 mg/mL of Pineapple and Southern Home cultivar extracts, respectively (Figure 4), by 72 h of exposure. In the three cancer cells tested, the muscadine grape extracts were found to be more effective in the A549 cells. Extracts (1000 ng/mL) of both of the cultivars reduced the cell survival to approximately 20% by 72 h of exposure (Figure 4).

### 3.3. Resveratrol Alone is Less Cytotoxic Than the Whole Berry Extract

Effect of resveratrol on the cancer cells was tested with different concentrations from 250 ng/mL to 50 µg/mL. The observations indicate that the pure resveratrol was effective only at higher concentrations (from 10 µg/mL) compared to the stilbene-rich muscadine grape extracts, where the active concentrations were less than 1 µg/mL. In HepG2 cells, pure resveratrol was able to reduce the number of surviving cells by 50% at 30 µg/mL concentration by 72 h (Figure 5a). At higher concentrations (50 µg/mL), cell numbers could be reduced by half in 48 h.

In HCC1806 cells, the better cytotoxicity was observed with 50 µg/mL of resveratrol by 72 h (Figure 5b). Though the lower concentrations tested showed cytotoxicity, their effect was not predominant in significantly reducing the cell numbers to half.

More than a 50% reduction in the A549 cell population was observed with 20 µg/mL resveratrol-treated cells by 48 h (Figure 5c). At higher concentrations (50 µg/mL), cell survival was reduced by one-fourth in the same time of exposure. By 72 h, only 20% of A549 cells survived in the 30, 40, and 50 µg/mL resveratrol-treated group (Figure 5c).

### 3.4. Synergistic Action of Stilbenes Induces Better Cellular Response Than Resveratrol Alone

The results of the cytotoxicity studies with HepG2, HCC1806, and A549 cells indicate that the stilbene-rich muscadine grape extracts show better cytotoxicity against the cancer cells than the resveratrol alone. The extracts were found to be at least 10-fold more effective in inducing cell death than the pure compound resveratrol. To ascertain this observation at the molecular level, semi-quantitative gene expression studies were performed with one of the representative cell lines, HepG2 cells treated with 1 µg/mL of extract and 50 µg/mL of pure resveratrol for 24 h. The list included some of the genes involved in tumor suppression (*Fas ligand*, *p53,* and *cas8*) and tumor progression (*Bcl2*, *EGFR*, *VEGF*, *cdk,* and *ybx1*). The list of genes studied and the primers used for PCR amplification are presented in Table 2.

Pure resveratrol was found to significantly (*p* < 0.005) reduce the expression levels of *VEGF* and *cdk,* while had slight reduction (*p* < 0.05) in the expression levels of *Bcl2* and *EGFR*. Treatment of HepG2 cells with resveratrol (50 µg/mL) for 24 h did not significantly alter the expression levels of y-box protein gene (*ybx1*) expression (Figure 6. Stilbene-rich extract of muscadine grape cultivar Pineapple significantly (*p* < 0.0005) reduced the expression of all of the tumor-promoting genes analyzed in this study. Cultivar Southern Home significantly reduced the expression levels of *VEGF* and *cdk,* but did not show any effect on *Bcl2*, *EGFR,* and *ybx1* (Figure 6).

Expression levels of tumor suppressor genes (*Fas ligand*, *p53,* and *cas8*) were significantly upregulated (*p* < 0.0005) by both Pineapple and Southern Home cultivars of muscadine grapes (Figure 7). While resveratrol alone too upregulated these genes, the effect was marginal and comparatively lesser than the muscadine grape extracts tested in this study.

## 4. Discussion

Fruits are a major source of bioactive phytochemicals, which are consumed without much processing. Anti-cancer effects of several muscadine grape cultivars have been reported earlier [6,26,27]. The most commonly studied genotypes are noble, carlos, and dixie red. While the majority of studies have used pomace, skin, and seed extracts, some have used juice and wine. The magnitude of cancer cell cytotoxicity responses varies with the muscadine varieties, parts used, and method of preparation. This can be attributed to the phytochemical content of the extract. In the current study, we used two of the commonly found varieties that produce black (Southern Home) and bronze (Pineapple) berries. The extraction method followed in this study extracted the major stilbenes present in muscadine grape berries. The varieties also showed variation in the composition of stilbenes tested. This study is the first report on the anti-cancer property of Southern Home and Pineapple cultivars of muscadine grapes. The secondary metabolite content of plants can greatly vary based on various factors, but certain cultivars of muscadine grapes, including the cultivars selected for this study, were found to consistently produce stilbenes [16].

Stilbenes are a family of plant secondary metabolites derived from the phenylpropanoid pathway and the acetate–malonate pathway [16]. Over the last two decades, stilbenes have gained popularity due to the health benefits associated with their consumption. However, resveratrol is the most studied and has been reported to be effective against various cancers, including breast, lung, colon, skin, prostate, ovarian, liver, oral cavities, thyroid, and leukemia [28]. Though plants contain multiple stilbenes, the health benefits of other stilbenes like piceid, viniferin, pterostilbene, and their combinations have not been focused on much.

In vitro studies have shown effective cytotoxic potential of resveratrol at concentrations up to 200 µM, while the physiological concentrations of resveratrol do not go beyond 50 nM [29]. This is due to the poor bioavailability of resveratrol and leads to differential anti-tumor effects in in vitro and in vivo. Higher doses of resveratrol were found to impair cell division and induce pro-apoptotic factors in normal cells [30]. The current study shows that the combination of stilbenes could be more effective than the individual resveratrol. The cytotoxic effect found with 50 µg/mL of resveratrol was comparable with 1 µg/mL of the stilbene-rich extract of the muscadine grape cultivars. Further, modulation of tumor suppressor and promotor gene expression was more predominant with the extract than the resveratrol alone.

Pterostilbene has been tested for its cytotoxicity against lung, breast, and hepatic cancer cells. The pterostilbene showed anti-proliferative activity (IC_50_) in MCF7 (breast cancer) and NCI H-460 (lung cancer) cell lines at 30.0 and 47.2 µM [31]. However, a combination of resveratrol (15 μM) and pterostilbene (5 μM) inhibited proliferation of HCC1806 cell lines [32]. Piceid was found to have anti-proliferative effects in intestinal epithelial Caco-2 cells in a concentration-dependent manner from 1 to 50 μM [33]. Studies have shown that the piceid is metabolized by cells to form resveratrol, which induces biological effects. However, Storniolo et al. [33] did not observe deglycosylation of piceid to resveratrol by Caco-2 cells. So, the cytotoxic effect was attributed to piceid in the medium. ε-viniferin has been reported to have a cytotoxic effect against various human cancers. Viniferin tested against melanoma cells (HT-144 and SKMEL-28) and healthy dermal fibroblast (HDF) cells showed an IC_50_ value at concentrations around 90 μM, while that of resveratrol was at >100 μM [34].

All of the major stilbenes tested in this study showed cytotoxic effect against various cancer cells. In fact, stilbenes like piceid performed better than resveratrol in in vitro studies. Nevertheless, the effective concentrations reported depends on various factors, but it is evident that the inhibitory concentrations of stilbenes differ with cancer cell type. Diseases like cancer are systemic, involving multifactorial etiology; in such conditions, treating them with one molecule and one target paradigm cannot be successful. Multicomponent drugs consist of several compounds that interact with multiple targets [35]. System biology understanding of multicomponent drugs indicates that they have superior efficacy and reduced toxicity. The optimal combinations of these stilbenes might have a better outcome in inducing cancer cell death.

Cancer cell lines have been widely used in research for understanding the biological mechanisms involved in cancer. The observations of the research in cancer cell lines can usually be extrapolated to in vivo human tumors and largely used in primary screening in cancer drug discovery. To date, several thousands of molecules have been tested positive for their anti-cancer effect against various types of cancer. However, the majority of them fail to reproduce the effect in vivo. So, further studies with the muscadine grape extracts will be required before using them for therapeutic purposes.

The mode of action of stilbenes is mainly through arresting mitosis, inhibiting telomerase activity, arresting DNA damage repair processes, inducing oxidative stress, activating mitophagy, and leading to apoptosis [36]. However, other additional effects, such as anti-inflammatory activity, and reduction in growth factors like *VEGF* and *PDGF* are also observed [28]. In the current study, stilbene-rich muscadine grape extracts of Southern Home and Pineapple cultivars were also found to induce apoptosis-related gene expression (*Bcl2* and *Cas8*). Though there was a difference in the magnitude of different genes expressions, the trend was found to be the same with both of the cultivars. The overall molecular process by which muscadine grape extracts tested induce cell death is summarized in Figure 8.

## 5. Conclusions

With the current understanding, it can be concluded that the synergistic action of stilbenes is more effective in killing the cancer cells than the resveratrol alone. The stilbenes alter the expression levels of tumor suppressor and tumor promoter genes to bring about cell death. The exact combination and proportion of stilbenes required for inducing cancer cell death requires further investigation.

## Figures and Tables

**Figure 1 biomedicines-07-00096-f001:**
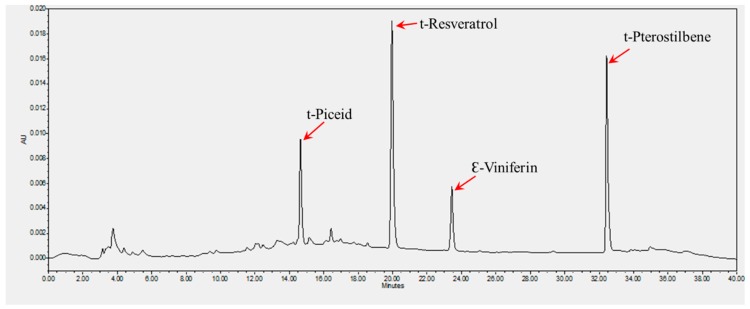
HPCL chromatogram of stilbene standards. The analytical column Luna RP C18 (4.6 × 250 mm; particle size, 5 μm) and guard cartridge (C18 4 × 3.0 mm) were used. The column temperature was maintained at 25 °C. Water and acetonitrile were used as solvents for gradient elution. The flow rate was set at 0.4 mL/min. UV absorbance detection was recorded at 285 nm. The arrow marks indicate the retention time of each of the stilbene compound.

**Figure 2 biomedicines-07-00096-f002:**
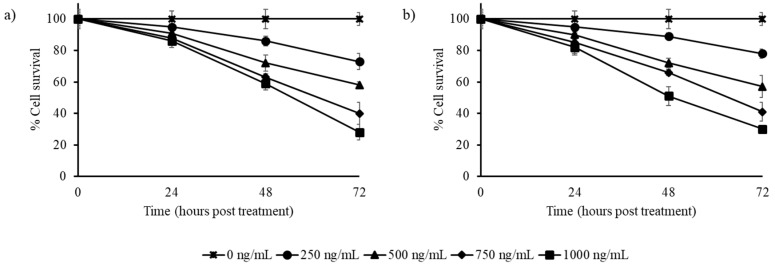
Effect of stilbene-rich muscadine grape extracts on the cytotoxicity of HepG2 cells; (**a**) Pineapple and (**b**) Southern Home cultivars. Cells were treated with different concentrations (0–1000 ng/mL) of stilbene-rich extracts from the berries of the two cultivars. The extracts induced differential cytotoxic response in HepG2 liver cancer cells. In < 48 h, about 50% of cells were killed by the stilbene-rich extracts of both of the cultivars. By 72 h of treatment, about 70–80% of cells were killed by the highest concentration (1 ng/mL) of the extracts tested.

**Figure 3 biomedicines-07-00096-f003:**
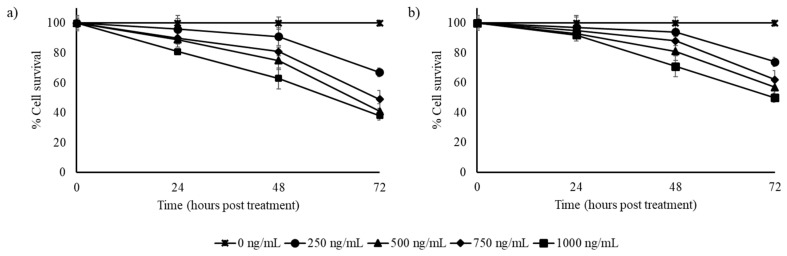
Effect of stilbene-rich muscadine grape extracts on the cytotoxicity of HCC1806 cells; (**a**) Pineapple and (**b**) Southern Home cultivars. Cells were treated with different concentrations (0–1000 ng/mL) of stilbene-rich extracts from the berries of the two cultivars. The extracts induced differential cytotoxic response in HCC1806 triple negative breast cancer cells. More than 50 h was required to kill 50% of cells by the highest concentration (1000 ng/mL) of the extracts tested.

**Figure 4 biomedicines-07-00096-f004:**
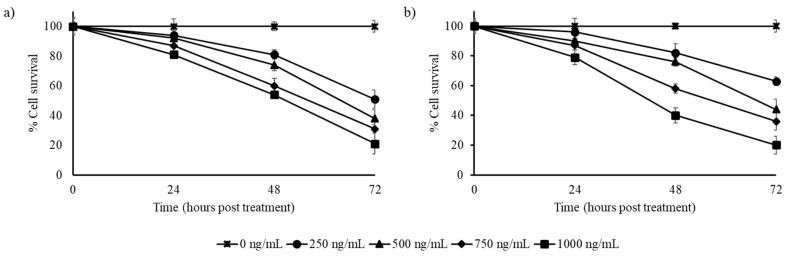
Effect of stilbene-rich muscadine grape extracts on the cytotoxicity of A549 cells; (**a**) Pineapple and (**b**) Southern Home cultivars. Cells were treated with different concentrations (0–1000 ng/mL) of stilbene-rich extracts from the berries of the two cultivars. Both extracts induced cell death in A549 cells. About 50% of cells were killed in < 30 h. By 72 h of treatment, 80% of cells were killed by the highest concentration (1 ng/mL) of the extract of both the cultivars tested.

**Figure 5 biomedicines-07-00096-f005:**
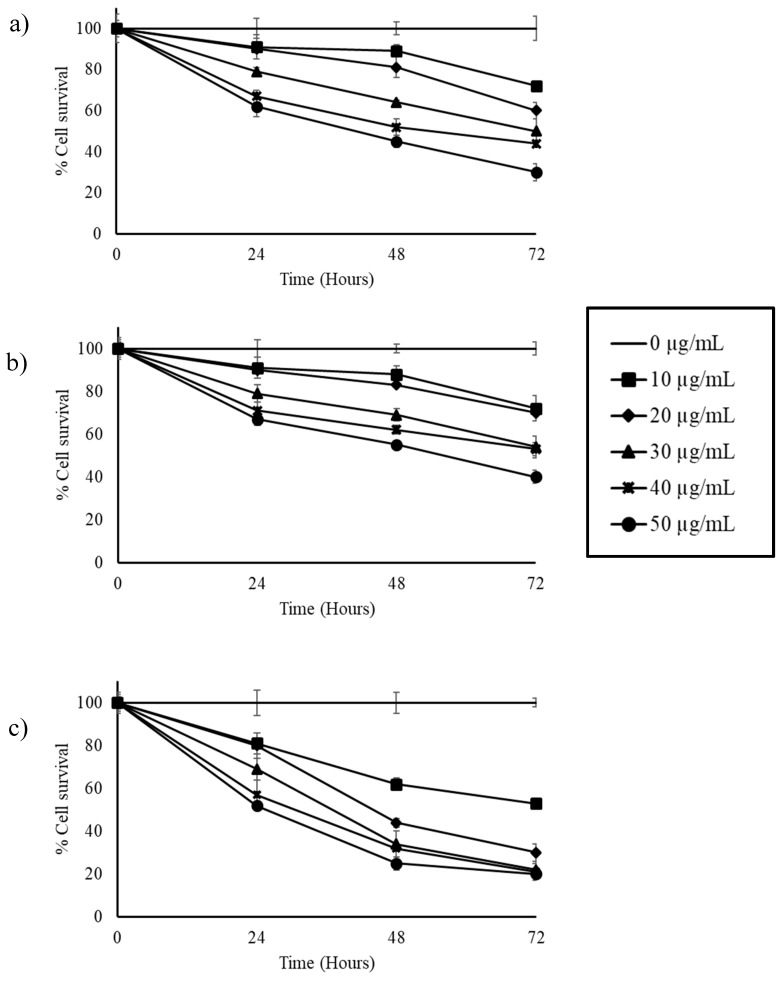
Effect of resveratrol on cytotoxicity of (**a**) HepG2, (**b**) HCC1806, and (**c**) A549 cells. Cells were treated with resveratrol at concentrations of 10–50 µg/mL. All concentrations tested showed cytotoxic activity against cancer cells. Differential response was observed with each of the cell lines tested. About 50% cell death could be observed with the highest concentration tested (50 µg/mL) in about 24 h treatment with resveratrol. In A549 cells, 30, 40, and 50 µg/mL showed similar maximum responses at 72 h.

**Figure 6 biomedicines-07-00096-f006:**
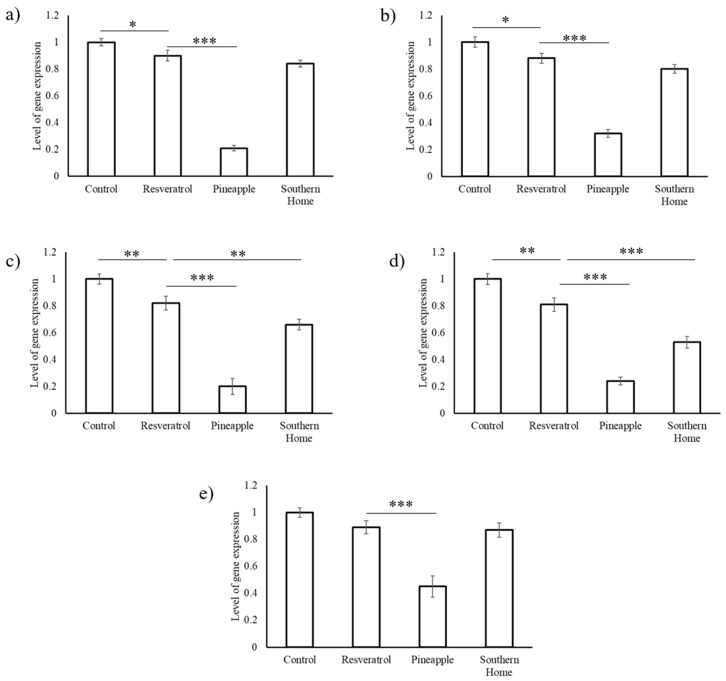
Effect of stilbene-rich muscadine grape extracts on the tumor-promoting genes in HepG2 cells. (**a**) *Bcl2,* (**b**) *EGFR,* (**c**) *VEGF,* (**d**) *cdk,* and (**e**) *ybx1*. Stilbene-rich extracts (1 µg/mL) muscadine grape cultivars, Pineapple and Southern Home, were found to significantly downregulate the expression of the tumor-promoting genes. Resveratrol (50 µg/mL) was found to have marginal repression of these genes, but was less effective than the berry extracts, even at a 50-fold higher dose. (* *p* < 0.05; ** *p* < 0.005; *** *p* < 0.0005).

**Figure 7 biomedicines-07-00096-f007:**
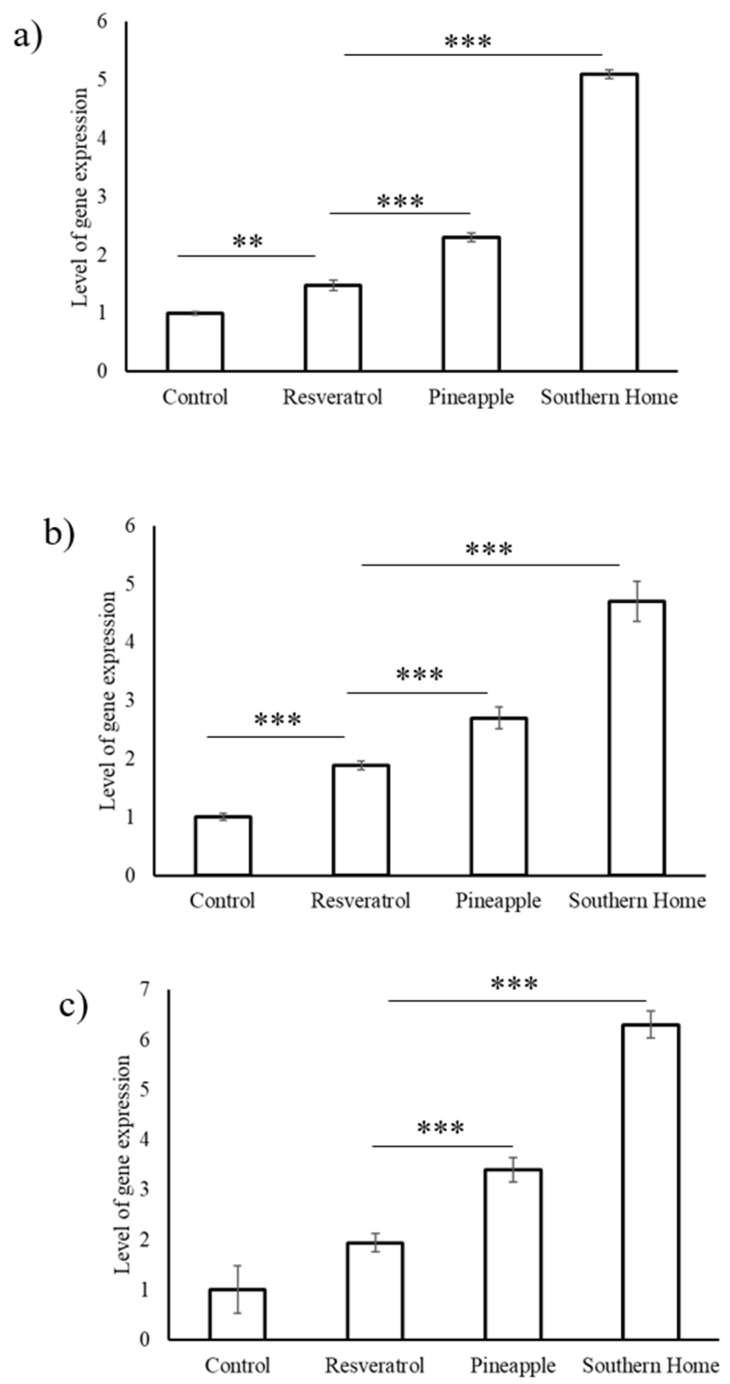
Effect of stilbene-rich muscadine grape extracts on the tumor-suppressing genes in HepG2 cells. (**a**) *Fas ligand,* (**b**) *p53,* and (**c**) *cas8*. Stilbene-rich extracts (1 µg/mL) muscadine grape cultivars, Pineapple and Southern Home, were found to significantly (*p* < 0.0005) activate the expression of the tumor suppressor genes. Resveratrol (50 µg/mL) was found to have moderate activation of these genes, but lesser than the berry extracts. (** *p* < 0.005; *** *p* < 0.0005).

**Figure 8 biomedicines-07-00096-f008:**
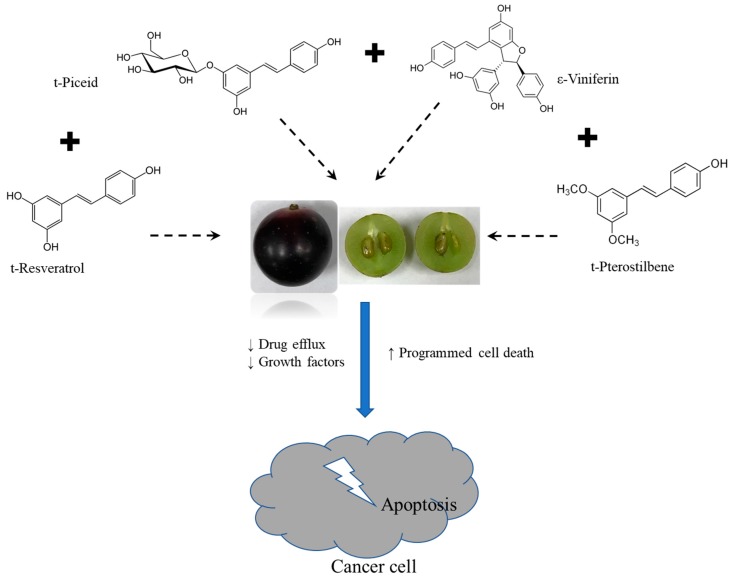
Figure showing the effect of stilbene-rich muscadine grape extract in cancer cells. The synergistic action of stilbenes present in muscadine grape berries reduces drug efflux, reduces cancer cell growth, and activates programmed cell death to kill the cancer cells. (↑ — upregulation; ↓ — downregulation)

**Table 1 biomedicines-07-00096-t001:** Quantification of stilbenes in the muscadine grape cultivars used in this study.

Cultivar	Berry Color	Stilbene Content (µg/g)			
		t-Piceid	t-Resveratrol	Ɛ-Viniferin	t-Pterostilbene
Pineapple	Bronze	279.75 ± 25	36.85 ± 1.15	24.35 ± 0.65	ND *
Southern Home	Black	480 ± 20	35 ± 1	1157.5 ± 22.5	4.5 ± 0.1

* ND: Not detected.

**Table 2 biomedicines-07-00096-t002:** Details of primers used in semi-quantitative gene expression.

Gene	Position	Primer Sequence (3′ to 5′)	Annealing Temperature	Amplicon Size (bp)	Reference
*GAPDH*	F	GACCACAGTCCATGCCATCA	60	450	[23]
R	TCCACCACCCTGTTGCTGTA		
*Bcl-2*	F	ATGTCCAGCCAGCTGCACCTGAC	60	319
R	GCAGAGTCTTCAGAGACAGCCAGG		
*Fas*	F	CAGGCTAACCCCACTCTATG	61	450
R	TGGGGGTGCATTAGGCCATT		
*Cas-8*	F	ACTTCAGACACCAGGCAGGGCT	62	500
R	GCCCCTGCATCCAAGTGTGTTC		
*ybx-1*	F	GACTGCCATAGAGAATAACCCCAG	62	496
R	CTCTCTAGGCTGTTTTGGGCGAGGA		
*EGFR*	F	GGAGCCTCTTACACCCAGTG	61	198	[24]
R	GCTTTCGGAGATGTTGCTTC		
*VEGF-A*	F	CACATAGGAGAGATGAGCTTC	60	100
R	CCGCCTCGGCTTGTCACAT		
*cdk1*	F	GGGTCAGCTCGCTACTCAAC	61	333	[25]
R	AAGTTTTTGACGTGGGATGC		
*p53*	F	TGTGGAGTATTTGGATGACA	58	550
R	GAACATGAGTTTTTTATGGC		
*β-Actin*	F	TGGGTCAGAAGGATTCCTATGT	60	276	[24]
R	CAGCCTGGATAGCAACGTACA

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
