# Peer review of "Synergistic Action of Stilbenes in Muscadine Grape Berry Extract Shows Better Cytotoxic Potential Against Cancer Cells Than Resveratrol Alone"

_biomedicines, 2019, doi:10.3390/biomedicines7040096_

Round 1
Reviewer 1 Report
Balasubramani et al. manuscript is dedicated to assessment of cytotoxic potential of stilbenes in muscadine grape extracts to cancer cells. Overall it is an interesting and valuable study, that can raise an interest at first in the basic breast cancer research field, focusing on mechanisms of action of stilbenes. As authors already mention in the discussion, the results are promising and could be potentially used for in vivo studies. However, it would be beneficial to asses stilbene cytotoxicity on normal cells.
The manuscript is well written; however, some minor revisions and English language check can be performed. Figures and tables are clear and understandable. Methods are described in depth. Gene expression changes were assessed by semi-quantitative PCR which is not the most precise approach. If possible, validation by real-time PCR would be beneficial to elaborate more on action mechanisms in future. Nevertheless, results shown in this study are promising. Perhaps the answers to the questions can be incorporated in the manuscript.
Minor revisions and questions
Questions:
Why the semi-quantitative PCR was performed only for HepG2, but not other cancer cell lines used for cytotoxicity assay?
How many stilbenoid extracts were obtained? If more than one extract from each cultivar, then which extract was used in cytotoxicity assay or the extracts were pooled together for cytotoxicity assay?
Experimental section 2.4.2. Total RNA extraction and cDNA synthesis
Page 3, line 123: Manufacturer or model of spectrophotometer should be included.
Results section 3.4.
Page 8: Gene names should be in Italica.
Author Response
Dear Professor,
Thanks a lot for giving a detailed review on our manuscript.
We completely agree with the suggestion of using real-time PCR for verification of gene expression. The primers used in this study have been indicated to be highly stringent and specific for the targets. The product size of the amplicons produced are above 175 to 500+ bp except for EGFR and VEGF. As you may be aware, higher amplicon size will cause dye saturation in real-time PCR and may not reflect the actual copy numbers, so, we have used semi-quantitative PCR. We are working on primers with similar stringency and lesser amplicon size to adopt real-time PCR in further studies.
Minor revisions and questions
Questions:
Why the semi-quantitative PCR was performed only for HepG2, but not other cancer cell lines used for cytotoxicity assay?
Answer:
As the extract showed a similar cytotoxic activity in all the three cell lines tested, we wanted to show the gene expression changes in one representative cell line. We used HepG2 cells because the genes targets identified in this study show transient expression in HepG2 cells, while on the A549 and HCC1806 cells there is uncertainty of transient expression of some of these genes. We have included a sentence indicating this in the results (line #227, page 8)
Question:
How many stilbenoid extracts were obtained? If more than one extract from each cultivar, then which extract was used in cytotoxicity assay or the extracts were pooled together for cytotoxicity assay?
Answer:
We have used only one stilbene rich extract in this study.
Question:
Experimental section 2.4.2. Total RNA extraction and cDNA synthesis
Answer:
Done. Heading font changed to italics.
Question:
Page 3, line 123: Manufacturer or model of spectrophotometer should be included.
Answer:
Done. Added the details of the manufacturer.
Question:
Results section 3.4.
Answer:
Done. Heading font changed to italics.
Question:
Page 8: Gene names should be in Italica.
Answer:
Done. Thanks for pointing out this. We have done this throughout the manuscript.
Reviewer 2 Report
The reviewed publication raises a very important scientific, functional and utilitarian problem. It is well known that resveratrol has extremely beneficial properties for the human health and body condition. It is used in various forms and also as an ingredient of many natural products. It often happens that mixtures containing a set of stilbenoids are characterized by higher activity than their individual components. There may be various reasons for this, including synergism of the action of individual components or significantly stronger activity of a component other than resveratrol.
The reviewed article also describes the above fact. However, its cause was explained in a somewhat unconvincing way. The research described in the paper is so broad and comprehensive and based on correct and respective assumptions, that it was possible to draw much more far-reaching conclusions from the obtained results than those quoted here.
However, considering the fact of the existence of many papers describing similar problems in relation to other natural products, it should be stated that the authors approached the examined issue very carefully and meticulously. The research conception is correctly put forward and their thesis are then systematically proven using the widest possible set of analytical methods.
The evaluated paper as a whole text meets all the criteria of the original research article prepared correctly for publication.
But there are a few minor editorial errors, which for the principle of “good reviewing practice” I list below:
- on line 30, the phrase "more than > 100" is a tautologies and probably should be removed the sign ">" (or the word "more than"),
- the title of the sub-chapter in line 71 should probably also have its numbering and in my opinion, it should be 2.3.1.
- in the caption of Fig. 5 and Fig. 6 the words “** 8” were incorrectly used instead of “***”,
Fig. 6 is described as "probably mode of action". In my opinion, it does not present how is the mode of action of these substances, although it Figure beautifully sums up all the work. I think that the present picture should be left but the title of them should be modified accordingly to the real content.
However, the content of Table 1 and any conclusions arising therefrom may be subject to substantive discussion. It is clearly seen here that in both analysed plant materials the content of resveratrol is low, so it is probably not the reason for the marked activity. Also, a component with a very diverse content between both plant materials cannot be the reason for quite similar activity. Despite a very broad discussion of the results obtained in Chapter 4, I lack in it sufficiently thorough analysis of the impact of the proportions of individual stilbene components in the assessed raw materials on the results obtained.
In addition, I believe that it would be a good idea to separate new Chapter 5. from actually existing long Chapter 4. titled “Discussion” of results, and present there clearly the conclusions drawn from this research.
Summarize my review, I believe that after minor corrections in the scope presented above, the article will be suitable for publication in the journal "Biomedicines".
Author Response
Dear Professor,
Thanks for a critical review of our manuscript.
- on line 30, the phrase "more than > 100" is a tautologies and probably should be removed the sign ">" (or the word "more than"),
Done. Thanks for pointing out this.
- the title of the sub-chapter in line 71 should probably also have its numbering and in my opinion, it should be 2.3.1.
Done.
- in the caption of Fig. 5 and Fig. 6 the words “** 8” were incorrectly used instead of “***”,
Done.
Fig. 6 is described as "probably mode of action". In my opinion, it does not present how is the mode of action of these substances, although it Figure beautifully sums up all the work. I think that the present picture should be left but the title of them should be modified accordingly to the real content.
Agree with the suggestion. The figure titled is now revised as “Figure showing effect of stilbene rich muscadine grape extract in cancer cells. The synergistic action of stilbenes present in muscadine grape berries reduce drug efflux, reduce cancer cell growth and activate programmed cell death to kill the cancer cells”
However, the content of Table 1 and any conclusions arising therefrom may be subject to substantive discussion. It is clearly seen here that in both analysed plant materials the content of resveratrol is low, so it is probably not the reason for the marked activity. Also, a component with a very diverse content between both plant materials cannot be the reason for quite similar activity. Despite a very broad discussion of the results obtained in Chapter 4, I lack in it sufficiently thorough analysis of the impact of the proportions of individual stilbene components in the assessed raw materials on the results obtained.
We completely agree with the observation. For detailed understanding, it is necessary to test the effect of individual stilbene compounds and in combinations in different concentrations. This will be a bigger study and will take longer time to accomplish.
In addition, I believe that it would be a good idea to separate new Chapter 5. from actually existing long Chapter 4. titled “Discussion” of results, and present there clearly the conclusions drawn from this research.
Done. We have now included a discussion section at the end of the manuscript (line # 327 to 332).